# Stochastic modeling of a gene regulatory network driving B cell development in germinal centers

**Alexey Koshkin**[1,2], **Ulysse Herbach**[3], **María Rodríguez Martínez**[4], **Olivier Gandrillon**[1,2]*, **Fabien Crauste**[5]*

**1** Inria Dracula, Villeurbanne, France, **2** Laboratory of Biology and Modelling of the Cell, Universite de Lyon, ENS de Lyon, Université Claude Bernard, CNRS UMR 5239, INSERM U1210, Lyon, France, **3** Université de Lorraine, CNRS, Inria, IECL, Nancy, France, **4** IBM Research Zurich, Zurich, Switzerland, **5** Université Paris Cité, CNRS, MAP5, Paris, France

* olivier.gandrillon@ens-lyon.fr (OG); fabien.crauste@math.cnrs.fr (FC)

**Data Availability Statement:** All biological data are available as Supporting information in the original publications: - Fig 2: Martinez et al. (2012) Quantitative modeling of the terminal differentiation

## Abstract

Germinal centers (GCs) are the key histological structures of the adaptive immune system, responsible for the development and selection of B cells producing high-affinity antibodies against antigens. Due to their level of complexity, unexpected malfunctioning may lead to a range of pathologies, including various malignant formations. One promising way to improve the understanding of malignant transformation is to study the underlying gene regulatory networks (GRNs) associated with cell development and differentiation. Evaluation and inference of the GRN structure from gene expression data is a challenging task in systems biology: recent achievements in single-cell (SC) transcriptomics allow the generation of SC gene expression data, which can be used to sharpen the knowledge on GRN structure. In order to understand whether a particular network of three key gene regulators (BCL6, IRF4, BLIMP1), influenced by two external stimuli signals (surface receptors BCR and CD40), is able to describe GC B cell differentiation, we used a stochastic model to fit SC transcriptomic data from a human lymphoid organ dataset. The model is defined mathematically as a piecewise-deterministic Markov process. We showed that after parameter tuning, the model qualitatively recapitulates mRNA distributions corresponding to GC and plasmablast stages of B cell differentiation. Thus, the model can assist in validating the GRN structure and, in the future, could lead to better understanding of the different types of dysfunction of the regulatory mechanisms.

## Introduction

Adaptive immune response is a complex mechanism, relying on B and T lymphocytes, which protects the organism against a range of pathogens. Crucial elements of adaptive immune response, the germinal centers (GCs) are the structures in lymphoid organs where activated naive B cells are expanded (in a dark zone, DZ) and selected (in a light zone, LZ) and can have multiple exit fates, such as antibody production (plasmablasts and plasma cells, PB_PC), long term storage of antigen information (memory B cells, MC), or death via apoptosis [1, 2].

of B cells and mechanisms of lymphomagenesis. Proc Natl Acad Sci 09(7): 2672-2677. - Figs 4&5: Milpied et al. (2018) Human germinal center transcriptional programs are desynchronized in B cell lymphoma. Nat Immunol 19(9): 1013-1024.

**Funding:** This work was supported by the COSMIC grant (www.cosmic-h2020.eu) which received funding from European Union's Horizon 2020 research and innovation program under the Marie Sklodowska-Curie grant agreement no. 765158. There was no additional external funding received for this study.

**Competing interests:** The authors have declared that no competing interests exist.

It is currently thought that B cell differentiation in GC is controlled by a small network of transcription factors (TFs) constituted by B-cell lymphoma 6 (BCL6), interferon regulatory factor 4 (IRF4) and PR domain zinc finger protein 1 (BLIMP1) [3]. BCL6 controls formation of GC, terminal differentiation of B cells and lymphomagenesis [4, 5]. BCL6 disturbance can be triggered by several mechanisms, including proteasome degradation by BCR, T-cell-mediated CD40-induced IRF4 repression of BCL6 [4, 6], or disruption of BCL6 autoregulation loop [4, 7]. Transcription factor IRF4 is involved in the termination of GC B cell differentiation, in immunoglobulin class switch recombination (CSR) and plasma cell development [8]. Impairment of IRF4 expression is tightly connected with the appearance of multiple malignancies [8]. BLIMP1 regulates pathways responsible for B cell lineage (e.g., PAX5) and GC proliferation and metabolism (e.g., MYC) [9, 10]. BLIMP1 is also involved in the induction of genes (e.g., XBP-1, ATF6, Ell2) facilitating antibody synthesis [11–13]. These three TFs interact, through various activation/inhibition processes: IRF4 represses BCL6 and activates BLIMP1 [14] (hence being essential for GC maturation and B cell differentiation into plasmablast), BLIMP1 and BCL6 mutually repress each other [15–18].

Martinez et al. [3] developed a deterministic kinetic ODE model capable of simulating normal and malignant GC exits using a GRN based on these three transcription factors. For the normal differentiation of GC B cells towards PB_PC stage, the kinetic ODE model fits microarray data at two steady-states: the first one associated with the GC stage of B cell differentiation (with high levels of BCL6 and low levels of IRF4 and BLIMP1), and the second one associated with PB_PC stage (with low levels of BCL6 and high levels of IRF4 and BLIMP1).

Recently, multiple protocols for SC RNA-seq data generation have been developed and used to answer various questions in biology [19, 20]. At the same time, different groups showed that gene transcription in eukaryotes is a discontinuous process and follows bursting kinetics [21–24]. Such results suggest that the stochastic nature of gene expression at the single cell (SC) level can be partly responsible for the phenotype variation in living organisms [25]. Thus, by gaining access to a stochastic behavior of gene expression, the SC viewpoint may lead to further improvement of the understanding of the biological systems and their variability.

Nevertheless, stochastic modeling of GRNs using SC gene expression data is still in its early stage [26, 27] and has never been studied for GC B cells. Here, we apply a particular class of stochastic models combining deterministic dynamics and random jumps, called piecewise-deterministic Markov processes (PDMPs) [28], to the description of GC B cell differentiation. It is a two-state model of gene expression introduced in [29] that allows a description of the system's dynamics at the promoter, transcription and translation levels for a given GRN. We apply this model to the GRN made of the three key genes, BCL6, IRF4 and BLIMP1, and simulate single B cell mRNA data [30]. We show that the model can qualitatively simulate the SC mRNA patterns for normal B cell differentiation at GC and PB_PC stages.

## Materials and methods

### Single-cell data

We used the B cell dataset from human lymphoid organs published by Milpied et al. [30]. The authors studied normal B cell subsets from germinal centers of the human spleen and tonsil and performed integrative SC analysis of gene expression. They used an adapted version of the integrative single-cell analysis protocol [31]. In short, the authors prepared cells for flow cytometry cell sorting. Then in every 96-well plate the authors sorted three to six ten-cell samples of the same phenotype as a single-cell. They performed multiplex qPCR analysis using the Biomark system (Fluidigm) with 96x96 microfluidic chips (fluidigm) and Taqmann assays (Thermofisher) [30]. They obtained results in the form of fixed fluorescence threshold to

derive *Ct* values. We used *Ct* values to derive Expression threshold (*Et*) values: $Et = 30 - Ct$. When there was an unreliably low or undetected expression ($Ct > 30$), *Et* was set to zero [30]. Using SC gene expression analysis of a panel of 91 preselected genes and pseudotime analysis (based on the cartesian coordinates of SC on the first and second principal components of the PCA), the authors separated GC DZ cells, GC LZ cells, memory cells and PB_PC cells.

Here we focused on three genes, BCL6, IRF4 and BLIMP1. We selected the SC gene expression values for BCL6, IRF4 and BLIMP1 for GC DZ cells (317 SC) and for PB_PC (104 SC) (see Fig 5). The experimental dataset includes at the GC B cell stage 30 cells with zero BCL6 mRNA amount, 292 cells with zero IRF4 mRNA amount and 292 cells with zero BLIMP1 mRNA amount. For the end of the B cell differentiation (PB_PC), there were 25 cells with zero BCL6 mRNA amount, 79 cells with zero IRF4 and 5 cells with zero BLIMP1 mRNA amount.

### Kinetic ODE model

Martinez et al. [3] derived an ODE model that simulates B cell differentiation from mature GC cells towards PB_PC. Dynamics of each protein (BCL6, IRF4 and BLIMP1) are defined by a production rate ($\mu$), a degradation rate ($\lambda$), a dissociation constant ($k$) and a maximum transcription rate ($\sigma$). Dynamics are described by System (1)–(3), where $p$, $b$ and $r$ account for proteins BLIMP1, BCL6 and IRF4, respectively:

$$\frac{dp}{dt} = \mu_p + \sigma_p \frac{k_b^2}{k_b^2 + b^2} + \sigma_p \frac{r^2}{k_r^2 + r^2} - \lambda_p p, \tag{1}$$

$$\frac{db}{dt} = \mu_b + \sigma_b \frac{k_p^2}{k_p^2 + p^2} \frac{k_b^2}{k_b^2 + b^2} \frac{k_r^2}{k_r^2 + r^2} - (\lambda_b + BCR)b, \tag{2}$$

$$\frac{dr}{dt} = \mu_r + \sigma_r \frac{r^2}{k_r^2 + r^2} + CD40 - \lambda_r r. \tag{3}$$

In this model, CD40 and BCR act as stimuli on genes: BCR temporary represses BCL6 and CD40 temporary activates IRF4.

### Stochastic model

The stochastic model that describes the coupled dynamics of gene *i* and the other genes of the GRN is defined by the series of equations:

$$
\begin{cases}
E_i(t) : 0 \xrightarrow{k_{\text{on},i}(P_1, P_2, P_3, Q_s)} 1, 1 \xrightarrow{k_{\text{off},i}(P_1, P_2, P_3, Q_s)} 0, \\
M_i'(t) = s_{0,i} E_i(t) - d_{0,i} M_i(t), \\
P_i'(t) = s_{1,i} M_i(t) - d_{1,i} P_i(t),
\end{cases} \tag{4}
$$

where $E_i(t)$, $M_i(t)$ and $P_i(t)$ are, respectively, the activation status of the promoter, the quantity of mRNA and the quantity of proteins of gene *i*, for $i \in \{1, 2, 3\}$. Each index *i* refers to one of the gene in the GRN, either BCL6, IRF4, or BLIMP1 (see Table 1). For $s \in \{BCR, CD40\}$, $Q_s$ accounts for external stimuli intensity.

For each gene *i*, System (4) is defined by the promoter state switching rates $k_{\text{on},i}$ ($\text{h}^{-1}$) and $k_{\text{off},i}$ ($\text{h}^{-1}$), by a degradation rate of mRNA ($d_{0,i}$, $\text{h}^{-1}$), a protein degradation rate ($d_{1,i}$, $\text{h}^{-1}$), a transcription rate ($s_{0,i}$, mRNA $\times$ $\text{h}^{-1}$), a translation rate ($s_{1,i}$, protein $\times$ mRNA$^{-1}$ $\times$ $\text{h}^{-1}$), and interaction parameters $\theta_{w,i}$ with either gene ($w = 1, 2, 3$) or stimulus ($w = BCR, CD40$).

**Table 1. Correspondence between gene names and model index.**

| Index | Gene/Stimulus |
|-------|---------------|
| 1 | BCL6 |
| 2 | IRF4 |
| 3 | BLIMP1 |

Interactions between genes are based on the assumption that $k_{\mathrm{on},i}$ and $k_{\mathrm{off},i}$ are functions of the proteins $P_1$, $P_2$, $P_3$ and stimuli $Q_s$. Parameter $k_{\mathrm{on},i}$ is given by:

$$k_{\mathrm{on},i}(P_1, P_2, P_3, Q_s) = \frac{k_{\mathrm{on},i}^{min} + k_{\mathrm{on},i}^{max}\beta_i\Phi_i(P_1, P_2, P_3, Q_s)}{1 + \beta_i\Phi_i(P_1, P_2, P_3, Q_s)} \tag{5}$$

where

$$\Phi_i(P_1, P_2, P_3, Q_s) = \prod_{s=\mathrm{BCR}}^{\mathrm{CD40}} \frac{1 + e^{\theta_{s,i}}Q_s/H_{s,i}}{1 + Q_s/H_{s,i}} \prod_{j=1}^{3} \frac{1 + e^{\theta_{j,i}}(P_j/H_{j,i})^\gamma}{1 + (P_j/H_{j,i})^\gamma}. \tag{6}$$

Parameter $H_{j,i}$ in (6) represents an interaction threshold for the protein $j$ on gene $i$ and $H_{s,i}$ an interaction threshold for stimulus $s$ on gene $i$, while in (5) $\beta_i$ is a scaling parameter. For defining $k_{\mathrm{off},i}$, all $\theta_{i,j}$ values must be replaced by $-\theta_{i,j}$ in (6). The structure of System (4)–(6) for the particular network considered in this paper is illustrated in Fig 1.

A detailed derivation of the model is presented in the supplementary material of [29]. Starting from a simple biochemical model of gene expression, the authors described higher-order interactions and took into consideration possible auto-activations. After normalization and simplification steps, Herbach et al. [29] and Bonnaffoux et al. [32] described the promoter switching rates $k_{\mathrm{on},i}$ and $k_{\mathrm{off},i}$ in the form of (5) and (6) by introducing the scaling parameter $\beta_i$. Following the approach in [32], the values of $\beta_i$ were computed when initializing the simulation, in order to set the values of parameters $k_{\mathrm{on},i}$ and $k_{\mathrm{off},i}$ to their initial values. Parameter $\gamma$ was set to a default value, equal to 2, and values of $k_{\mathrm{on},i}^{max}$ and $k_{\mathrm{off},i}^{max}$ were estimated by the method of moments and bootstrapping as previously described by Bonnaffoux et al. [32]. These parameters were no longer modified through this study.

It can be noted that the promoter state evolution of gene $i$ between times $t$ and $t + \delta t$ in System (4)–(6) is defined, for small $\delta t$, as a Bernoulli-distributed random variable [29, 32]:

$$E_i(t + \delta t) \sim \mathrm{Bernoulli}(\pi_i(t)),$$

where probability $\pi_i(t)$, derived by solving the master equation [29, 33], is given by

$$\pi_i(t) = E_i(t)e^{-\delta t(k_{\mathrm{on},i}+k_{\mathrm{off},i})} + \frac{k_{\mathrm{on},i}}{k_{\mathrm{on},i} + k_{\mathrm{off},i}}\left(1 - e^{-\delta t(k_{\mathrm{on},i}+k_{\mathrm{off},i})}\right).$$

It follows that the promoter state of gene $i$ averages to $k_{\mathrm{on},i}/(k_{\mathrm{on},i} + k_{\mathrm{off},i})$ in the fast promoter regime ($k_{\mathrm{on},i} + k_{\mathrm{off},i} \gg 1/\delta t$). This quantity will be used to reduce System (4)–(6) into an ordinary differential equation (ODE) system in Section.

## Simulating the stochastic model

During B cell differentiation in GC, B cells first receive BCR signal, through follicular dendritic cells interaction, that represses BCL6. Then, B cells integrate CD40 signals, through T follicular helper, that activate IRF4 [3, 6, 34].

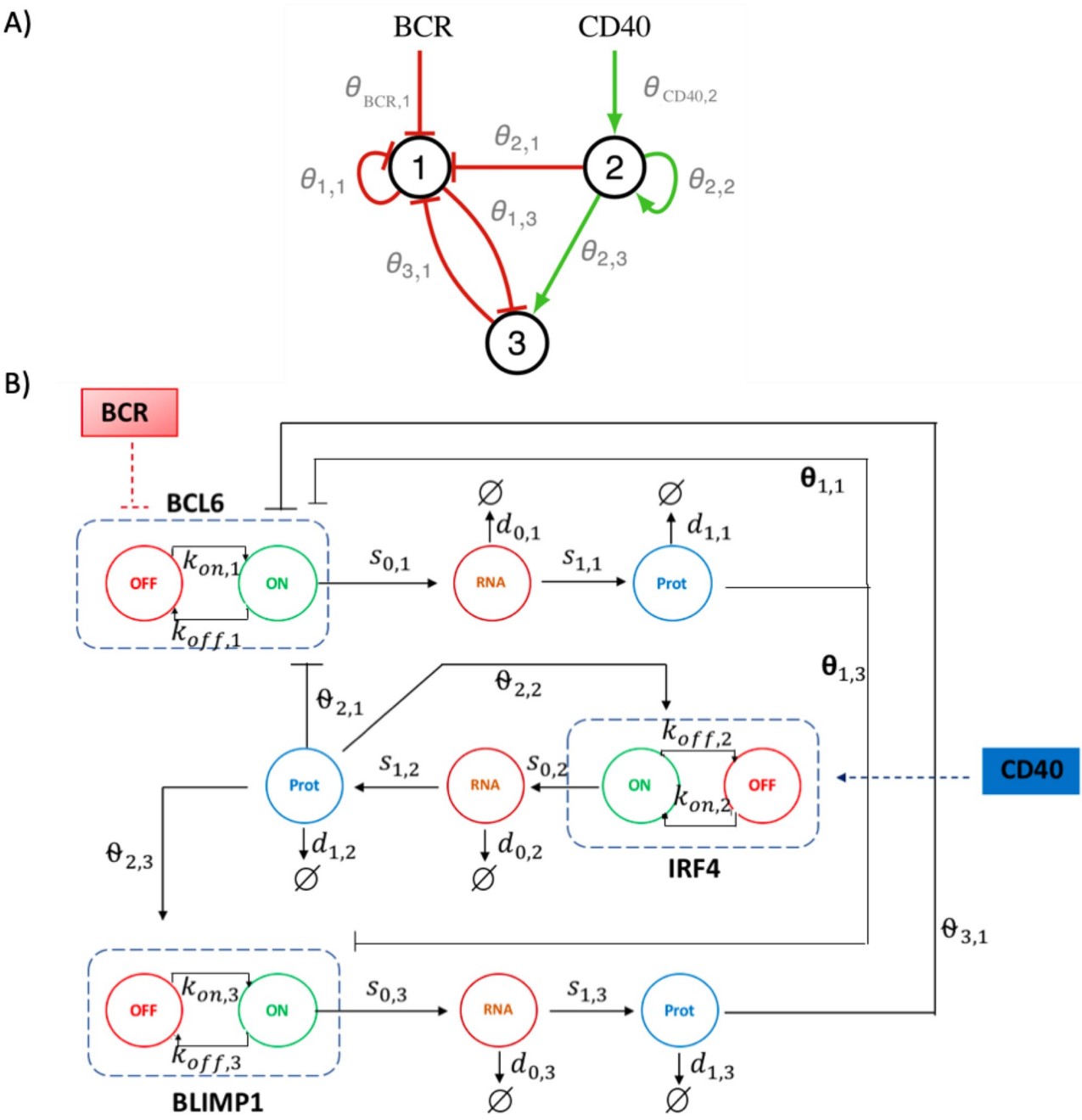

**Fig 1. Schematic representations of the GRN and the stochastic model.** A) Schematic representation of the three-gene GRN involved in B cell differentiation. It consists of BCL6 (gene 1), IRF4 (gene 2) and BLIMP1 (gene 3), and with stimuli BCR and CD40 acting on the network. The interaction $j \rightarrow i$ between a regulating protein $j$ and a target gene $i$ is represented by the interaction parameter $\theta_{j,i}$. B) Schematic representation of the associated stochastic model. A gene is represented by its promoter state (dashed rectangle), which can switch randomly from on to off (and vice versa), with rates $k_{\mathrm{on},i}$ ($k_{\mathrm{off},i}$). When promoter state is on, mRNA molecules are continuously produced at $s_{0,i}$ rate. Proteins are constantly translated from mRNA at $s_{1,i}$ rate. Parameters $d_{0,i}$ and $d_{1,i}$ are degradation rates of mRNA and proteins. The interaction between a regulator gene $j$ and a target gene $i$ is defined by the dependence of both $k_{\mathrm{on},i}$ and $k_{\mathrm{off},i}$ on the protein level $P_j$ and the interaction parameter $\theta_{j,i}$. IRF4 gene exhibits an autoactivation loop ($\theta_{2,2}$). BCL6 gene exhibits an autorepression loop ($\theta_{1,1}$). Additionally, two external stimuli, BCR and CD40, act on the GRN.

In order to simulate these interactions, we assumed that BCR was acting on BCL6 from 0h until 25h, and CD40 was acting on IRF4 from until 61h. Stimuli were implemented in three steps: first a linear increase ($t_{\mathrm{BCR}} \in [0.5h; 1.5h]$; $t_{\mathrm{CD40}} \in [35h; 36h]$), then a stable stimulus ($t_{\mathrm{BCR}} \in [1.5h; 24h]$; $t_{\mathrm{CD40}} \in [36h; 60h]$), finally a linear decrease ($t_{\mathrm{BCR}} \in [24h; 25h]$; $t_{\mathrm{CD40}} \in [60h; 61h]$) (see S1 Fig).

In all simulations, the system evolves for 500h so it can reach a steady state before applying the stimuli (at time $t = 0h$). After the first stimulus (BCR) is applied, the system is simulated for an additional 500h. For each simulation, the amounts of mRNA counts have been collected every 0.5h.

Values of parameters defining the stochastic system (4)–(6) are given in Tables 2 to 5.

**Table 2. Parameter set of the stochastic model (4)–(6) and reduced model (9) that are the same in all versions.**

| Parameter | Version I, II, III |
|---|---|
| $H_{1,2}$ | 1 |
| $H_{3,2}$ | 1 |
| $H_{3,3}$ | 1 |
| $\theta_{1,1}$ | -0.2 |
| $\theta_{1,2}$ | 0 |
| $\theta_{3,2}$ | 0 |
| $\theta_{1,3}$ | -1 |
| $\theta_{3,3}$ | 0 |
| $s_{0,\mathrm{BCL6}}$ | 100 |
| $d_{0,\mathrm{BCL6}}$ | 0.05 |
| $d_{0,\mathrm{IRF4}}$ | 0.05 |
| $s_{1,\mathrm{BCL6}}$ | 100 |
| $s_{1,\mathrm{IRF4}}$ | 160 |
| $s_{1,\mathrm{BLIMP1}}$ | 40 |
| $d_{1,\mathrm{BCL6}}$ | 0.138 |
| $d_{1,\mathrm{IRF4}}$ | 0.173 |
| $d_{1,\mathrm{BLIMP1}}$ | 0.173 |
| $k_{\mathrm{off,init,BCL6}}$ | 1 |
| $k_{\mathrm{off,init,IRF4}}$ | 1 |
| $k_{\mathrm{off,init,BLIMP1}}$ | 1 |

Version I—initial parameter set. Version II—parameter set obtained from the automatized approach. Version III—parameter set obtained after semi-manual tuning. Parameters are defined in the text.

**Table 3. Parameter set of the stochastic model (4)–(6) and reduced model (9) that are different between all versions.**

| Parameter | Version I | Version II | Version III |
|---|---|---|---|
| $H_{1,3}$ | 0.1 | 1 | 0.01 |
| $H_{\mathrm{BCR},1}$ | 0.01 | 1 | 0.001 |
| $H_{\mathrm{CD40},2}$ | 1 | 0.001 | 1 |

Version I—initial parameter set. Version II—parameter set obtained from the automatized approach. Version III—parameter set obtained after semi-manual tuning. Parameters are defined in the text.

**Table 4. Parameter set of the stochastic model (4)–(6) and reduced model (9) that are equal between versions I and II.**

| Parameter | Version I, II | Version III |
|---|---|---|
| $H_{2,2}$ | 0.01 | 0.1 |
| $\theta_{2,1}$ | -100 | -50 |
| $\theta_{2,2}$ | 5 | 11 |
| $\theta_{2,3}$ | 40 | 50 |
| $\theta_{3,1}$ | -20 | -0.5 |
| $\theta_{\mathrm{BCR},1}$ | -20 | -200 |
| $\theta_{\mathrm{CD40},2}$ | 40 | 10 |
| $d_{0,\mathrm{BLIMP1}}$ | 0.1733 | 0.007 |
| $s_{0,\mathrm{IRF4}}$ | 1 | 2.1 |
| $s_{0,\mathrm{BLIMP1}}$ | 1 | 100 |

Version I—initial parameter set. Version II—parameter set obtained from the automatized approach. Version III—parameter set obtained after semi-manual tuning. Parameters are defined in the text.

**Table 5. Parameter set of the stochastic model (4)–(6) and reduced model (9) that are equal between versions II and III.**

| Parameter | Version I | Version II, III |
|---|---|---|
| $H_{1,1}$ | 1 | 0.1 |
| $H_{2,1}$ | 0.1 | 0.01 |
| $H_{2,3}$ | 0.01 | 0.1 |
| $H_{3,1}$ | 1 | 0.01 |
| $k_{\mathrm{on,init,BCL6}}$ | 0.1 | 0.15 |
| $k_{\mathrm{on,init,IRF4}}$ | 0.0017 | 0.007 |
| $k_{\mathrm{on,init,BLIMP1}}$ | 0.1 | 0.001 |

Version I—initial parameter set. Version II—parameter set obtained from the automatized approach. Version III—parameter set obtained after semi-manual tuning. Parameters are defined in the text.

## Model execution in a computational center

All models were established as part of the WASABI pipeline [32] and were implemented in Python 3. All computations were performed using the computational center of IN2P3 (Villeurbanne/France).

## Tuning of the PDMP model

**Parameters estimation for the ODE-reduced model.** In Section, we use a reduced, deterministic version of System (4)–(6), namely System (9). Initial guess of each parameter has been chosen randomly in the same order of magnitude as in Bonnaffoux et al. [32]. Specifically, the initial value of $k_{\mathrm{on}}$ for IRF4 ($k_{\mathrm{on,init,IRF4}}$) has been estimated by comparison with values of the kinetic model from Martinez et al. [3]. Initial values of $k_{\mathrm{on}}$ for BCL6 and BLIMP1 were selected in the same order of magnitude as $k_{\mathrm{on,init,IRF4}}$.

**Estimation of the parameters for the stochastic model: Automatized approach.** After we have established the parameters for the reduced model (9), and we have shown that (9) has two steady states, we used these values as initial guess for the stochastic model (4)–(6). The

goal was then to further tune parameter values so the stochastic model (4)–(6) fits the experimental SC data.

We investigated a possible effect of $H_{w,i}$ values ($w$ = 1, 2, 3, BCR, CD40), $\theta_{j,i}$ values and $k_{\text{on,init}}$ values on the quality of the fitting (for each parameter combination, simulation was performed for 200 SC).

First, let us mention that, based on the network depicted in Fig 1, there is no influence of BCL6 on IRF4, of BLIMP1 on IRF4 and there is no self-activity of BLIMP1 on itself, so parameters $\theta_{1,2}$, $\theta_{3,2}$ and $\theta_{3,3}$ are set to 0 while parameters $H_{1,2}$, $H_{3,2}$ and $H_{3,3}$ are set to 1. Also, BCR acts only on BCL6 and CD40 on IRF4, so $\theta_{BCR,2} = \theta_{BCR,3} = 0$, $\theta_{CD40,1} = \theta_{CD40,3} = 0$, and only parameters $\theta_{BCR,1}$ and $\theta_{CD40,2}$ are non-zero.

We tested the values of interaction threshold $H_{1,1}$, $H_{1,3}$, $H_{2,1}$, $H_{2,3}$ and $H_{3,1}$ within the set {0.01, 0.1, 1}, and the set {0.0001, 0.001, 0.1, 1, 100} for $H_{2,2}$, $H_{BCR,1}$, $H_{CD40,2}$. We also tested the values of $\theta_{1,1}$, $\theta_{1,3}$, $\theta_{2,1}$, $\theta_{2,3}$ and $\theta_{3,1}$ by multiplying by a factor $f_\theta \in \{1, 5\}$, and by multiplying by a factor $f_\theta \in \{1, 10\}$ for $\theta_{2,2}$, $\theta_{BCR,1}$, $\theta_{CD40,2}$.

In total we tested two different values of $\theta_{j,i}$ for 5 interactions ($\theta_{1,1}$, $\theta_{1,3}$, $\theta_{2,1}$, $\theta_{2,3}$, $\theta_{3,1}$), 2 values of $\theta_{j,i}$ for 3 interactions ($\theta_{2,2}$, $\theta_{BCR,1}$, $\theta_{CD40,2}$), 3 values of $H_{j,i}$ for 5 interactions ($H_{1,1}$, $H_{1,3}$, $H_{2,1}$, $H_{2,3}$, $H_{3,1}$), and 5 values of $H_{j,i}$ for 3 interactions ($H_{BCR,1}$, $H_{CD40,2}$, $H_{2,2}$), generating $2^5 \times 2^3 \times 3^5 \times 5^3 \approx 7.8 \times 10^6$ combinations of parameters. Parameters that do not appear in the previous list have not been tested.

During this automatized tuning procedure, we selected a set of parameter values that allows the system to provide the best fit of the experimental mRNA values for BCL6, IRF4 and BLIMP1 at the GC stage, based on a quality-of-fit criterion. This criterion was defined as a comparison between the average model-derived values ($\Upsilon$) and the average experimental values ($\Omega$), with an objective function ($OF$) to minimize for the set of genes $G = \{BCL6, IRF4, BLIMP1\}$ and stages $ST = \{GC, PB\_PC\}$ defined by

$$OF = \sum_{\delta'=1}^{|G|}\sum_{\delta''=1}^{|ST|} \left| \frac{\Omega_{\delta',\delta''} - \Upsilon_{\delta',\delta''}}{\Omega_{\delta',\delta''}} \right|. \tag{7}$$

The quality-of-fit criterion is then

$$\min_{PS} OF, \tag{8}$$

where $PS$ is the set of parameter values from Tables 2 to 5.

**Estimation of the parameters for the stochastic model: Semi-manual tuning.** The automatized estimation procedure was followed by a semi-manual tuning of the parameters of the stochastic model (4)–(6) to improve the quality of the fit.

Values of candidate parameters have been tested in an interval of interest and the rest of the parameter values have been fixed at this stage. After model execution, model-simulated SC values of gene expression were collected. Then we selected the values of the parameters that provided the best qualitative fitting (see Eq (8)) of the experimental SC data. Ranges of tested values are summarised in Table 6.

## Evaluation of model variability using Kantorovich distance

To compare distributions and to evaluate model variability, we used the Kantorovich distance (KD, particular case of Wasserstein distance, WD), as defined by Baba et al. [35] and implemented in Python 3 by Bonnaffoux et al. [32].

**Table 6. Parameters tested during the semi-manual tuning of the stochastic model.**

| Parameter | Definition | Tested values | Selected value |
|---|---|---|---|
| $\theta_{1,1}$ | Interaction parameter | $[-200; -10^{-2}]$ | -0.2 |
| $\theta_{1,3}$ | Interaction parameter | $[-200; -0.1]$ | -1 |
| $\theta_{2,1}$ | Interaction parameter | $[-200; -10^{-2}]$ | -50 |
| $\theta_{2,2}$ | Interaction parameter | $[0.1; 200]$ | 11 |
| $\theta_{2,3}$ | Interaction parameter | $[0.1; 200]$ | 50 |
| $\theta_{3,1}$ | Interaction parameter | $[-200; -10^{-2}]$ | -0.5 |
| $\theta_{BCR,1}$ | Interaction parameter | $[-200; -0.1]$ | -200 |
| $\theta_{CD40,2}$ | Interaction parameter | $[0.1; 200]$ | 10 |
| $s_{0,BCL6}$ | Transcription rate | $[0.1; 625]$ | 100 |
| $s_{0,IRF4}$ | Transcription rate | $[0.1; 625]$ | 2.1 |
| $s_{0,BLIMP1}$ | Transcription rate | $[0.1; 625]$ | 100 |
| $d_{0,BCL6}$ | Degradation rate of mRNA | $[10^{-3}; 10]$ | 0.05 |
| $d_{0,IRF4}$ | Degradation rate of mRNA | $[10^{-3}; 10]$ | 0.05 |
| $d_{0,BLIMP1}$ | Degradation rate of mRNA | $[10^{-3}; 10]$ | 0.007 |
| $s_{1,BCL6}$ | Translation rate | $[1; 1000]$ | 100 |
| $s_{1,IRF4}$ | Translation rate | $[1; 1000]$ | 160 |
| $s_{1,BLIMP1}$ | Translation rate | $[1; 1000]$ | 40 |
| $d_{1,BCL6}$ | Degradation rate of protein | $[0.1; 10]$ | 0.138 |
| $d_{1,IRF4}$ | Degradation rate of protein | $[0.1; 10]$ | 0.173 |
| $d_{1,BLIMP1}$ | Degradation rate of protein | $[0.1; 10]$ | 0.173 |
| $k_{on,init,BCL6}$ | Initial value of $k_{on,BCL6}$ | $[10^{-5}; 10]$ | 0.15 |
| $k_{on,init,IRF4}$ | Initial value of $k_{on,IRF4}$ | $[10^{-5}; 10]$ | 0.007 |
| $k_{on,init,BLIMP1}$ | Initial value of $k_{on,BLIMP1}$ | $[10^{-5}; 10]$ | 0.001 |

Consider two discrete distributions $p$ and $q$, defined on $N$ bins of equal sizes, and denote by $x_k$ the center of the $k$-th bin. Then the Kantorovich distance between $p$ and $q$ is given by

$$KD = \sum_{n=1}^{N} \left| \sum_{k=1}^{n} p(x_k) - \sum_{k=1}^{n} q(x_k) \right|.$$

We chose WD because it suggested to be preferable over alternative methods such as Kullback-Leibler (KL) divergence or Jensen-Shannon (JS) divergence [36]. More specifically, WD does not require that distributions belong to the same probability space. At the same time, WD is more tractable and has higher performance compared to KL divergence [37]. JS divergence, in turn, does not provide a gradient for the distributions of non-overlapping domains, compared to WD [36]. Also, because WD is a metric and accounts both for the "cost" for the transfer (distance) and "the number of counts" to transfer, we selected its 1D case of WD (Kantorovich Distance, KD) for comparison of discrete experimental distributions versus model-derived distributions [38].

## Results

### Reduced model

In [3], Martinez et al. applied the kinetic ODE model (1)-(3) to the BCL6-IRF4-BLIMP1 GRN associated with B cell differentiation and successfully simulated GC B cell dynamics based on microarray data. Before using the complex, stochastic model (4)–(6) to fit SC data, we

considered a reduced version of System (4)–(6) that can be compared to model (1)–(3), hence providing an initial guess for a key parameter of the model.

Since model (1)–(3) is deterministic, it is necessary to simplify the stochastic model (4)–(6) to perform a comparison of both models dynamics. We assume, in this section, that the stochastic process $E(t)$ (promoter status) in (4)–(6) equals its mean value, $\langle E(t) \rangle$, given by $k_{on}/(k_{on} + k_{off})$. System (4)–(6) then reduces to

$$
\begin{cases}
\langle E(t) \rangle &=& \dfrac{k_{on}(t)}{k_{on}(t) + k_{off}(t)}, \\[2mm]
\dfrac{dM}{dt} &=& s_0 \langle E(t) \rangle - d_0 M(t), \\[2mm]
\dfrac{dP}{dt} &=& s_1 M(t) - d_1 P(t).
\end{cases}
\tag{9}
$$

Comparing mathematical formulations of systems (1)–(3) and (9), one can see that it is possible to identifiy an initial value of the promoter state $E(t)$ for IRF4 gene in System (9) that will correspond to GC differentiation stage (S1 File). Indeed, after rewriting System (9) in terms of System (1)–(3), we obtained the candidate value of $k_{on,init,IRF4} = 1.7 \times 10^{-3}$. Using this value of $k_{on,init,IRF4}$, System (9) successfully simulates two steady states for IRF4, i.e. it recapitulates the qualitative dynamics of System (1)–(3) (Fig 2).

Before application of BCR and CD40 stimuli, the system is at a steady state (simulating GC B cell stage) that corresponds to a low amount of IRF4 and BLIMP1 and a high amount of BCL6 mRNA molecules. After application of both stimuli, the system has transitioned to a second steady state that corresponds to a high number of IRF4 and BLIMP1 mRNA molecules and a low number of BCL6 mRNA molecules. However, it can be noted that for the current parameter set (Tables 2–5, version I), System (9) underestimates the amount of IRF4 mRNA at both steady states (Fig 2).

Dynamics of System (9) shows the existence of two steady-states for the parameter set from Tables 2–5, version I. Notably, if we test a random value of $k_{on,init,IRF4}$ in combination with the parameters from Tables 2–5, version I (S1 Table), System (9) has only one steady-state (S2 Fig). To our knowledge, there may be more than one set of parameter values associated with two steady states of System (9).

We showed that for the parameter set from Tables 2–5, version I, the reduced model (9) is capable to qualitatively recapitulating the expected behavior of GC B cell differentiation GRN (Fig 2). Due to the stochastic nature of gene expression, we are hereafter interested in evaluating how stochastic system (4)–(6) is capable of simulating this stochastic behavior in B cell differentiation in GC and recapitulates the shape of the mRNA distributions from the experimental SC dataset.

## Stochastic modeling of B cell differentiation

**Assessing the variability of the stochastic model.** Due to the stochastic nature of the stochastic system (4)–(6), it is important to first evaluate the variability of the model-generated SC data, that is of model's outputs. Indeed, when one repeatedly simulates a finite number of cells from the stochastic system (4)–(6) for the same parameter value set (Tables 2–5, version I), the resulting model-derived empirical distributions are slightly different between each run due to the stochasticity of the model. We investigated how strongly shapes of distributions of simulated SC mRNA molecules vary for different executions of model (4)–(6).

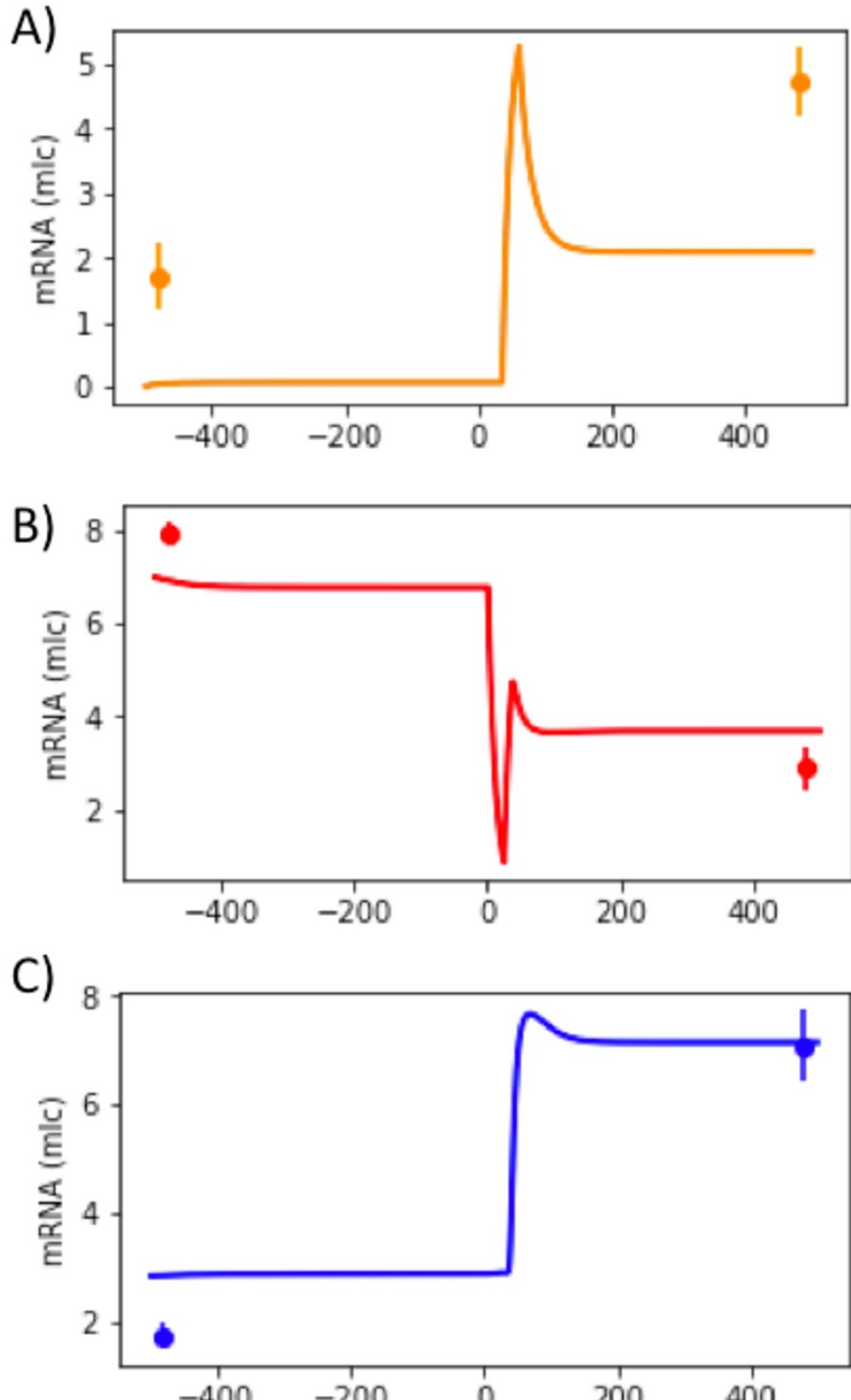

**Fig 2. Temporal evolution of mRNA counts generated by the reduced model (9).** Temporal evolutions of IRF4 (A), BCL6 (B) and BLIMP1 (C) (see Fig 1). BCR stimulus was applied from 0h until 25h and CD40 stimulus from 35h until 61h. Microarray gene expression dataset from GEO accession no. GSE12195 was used to estimate model's parameters (see Tables 2 to 5, version I) and are shown as dots with error bars.

We evaluated the level of variability of model (4)–(6) using the Kantorovich distance (KD, see Section). We simulated 200 datasets, each containing 200 single cells, of System (4)–(6) with a fixed parameter set (see Tables 2–5, version I). We estimated the KD between pairs of simulated datasets (mRNA counts for three genes at GC and PB_PC stages for 200 simulated cells), and obtained a distribution of all KD that we call the model-to-model (m-t-m) distribution (Fig 3). Shapes of m-t-m distributions are different for each gene and stage of differentiation. For instance, for BLIMP1, long tails are observed. We can also notice that the mean value of IRF4 at GC stage is low compared to other genes.

In order to get a more accurate evaluation of the variability in model's outputs, we plotted distributions of the number of mRNA molecules (model's outputs) for each node of the GRN with the highest m-t-m distribution at both GC and PB_PC stages (Fig 4). Qualitatively, no difference is detected in the shapes of model-generated distributions. For all 6 nodes, the shapes of distributions are remarkably similar.

These results suggest that it may be sufficient to perform parameter tuning of the stochastic model (4)–(6) using only one simulation run for each parameter value set.

**Initial estimation step based on an automatized approach.** Variability of the stochastic model being assessed, and comparison of experimental data and a single model's output in order to assess their closeness being validated, we now focus on the estimation of parameter values. Model (4)–(6) comprises 40 parameters, so we first apply a straightforward strategy, that we call automatized approach, which consists in solving the stochastic system (4)–(6) for a number of fixed parameter values and selecting the set of parameter values associated with the best fit (see Section Estimation of the parameters for the stochastic model: Automatized approach) of experimental data [30].

Approximately $8 \times 10^6$ combinations of parameter values have been tested (see Section Estimation of the parameters for the stochastic model: Automatized approach), then the best set of parameter values has been selected based on the quality of BCL6, IRF4 and BLIMP1 fitting at the GC and PB_PC stages (Eqs (7) and (8)).

Numbers of mRNA molecules estimated by the stochastic model (4)–(6) are in a similar range of magnitude as the experimental SC data (S3 Fig). However, the selected parameter values (Tables 2–5, version II) generate model-derived mRNA distributions that have sufficient overlap with experimental data for GC stage but insufficient overlap for PB_PC stage (S3 Fig). Indeed, distributions of numbers of mRNA molecules at PB_PC stage mostly underestimate the experimental SC data (S3B, S3D and S3F Fig).

Implementing an automatized approach for estimating parameter values helped to establish a set of parameter values that allows System (4)–(6) to correctly estimate the number of mRNA molecules for 3 out of 6 nodes of the GRN. In order to improve the quality of the fit, a more directed and sensitive tuning of the parameter set is then performed (see Section Estimation of the parameters for the stochastic model: Semi-manual tuning).

**Generation of simulated distributions of mRNA counts describing B cell differentiation.** Due to the complexity of the stochastic model (4)–(6) that is made of 40 parameters, it is important to identifiy which parameters should be targeted to improve the quality of fit. To do so, we rely on the properties of the GRN (Fig 1A). Thanks to the topological structure of the BCL6-IRF4-BLIMP1 GRN, where IRF4 activates BLIMP1 and autoactivates itself, we hypothesize that System (4)–(6) underestimates the experimental SC data at the PB_PC stage due to low values of the parameters responsible for IRF4 autoactivation ($\theta_{2,2}$, and to a lesser extent $s_{0,IRF4}$) and BLIMP1 activation by IRF4 ($\theta_{2,3}$). Further, we improved the quality of the fit, in particular of BLIMP1 distribution, by focusing on BLIMP1-related interaction parameters ($\theta_{1,3}, \theta_{3,1}$).

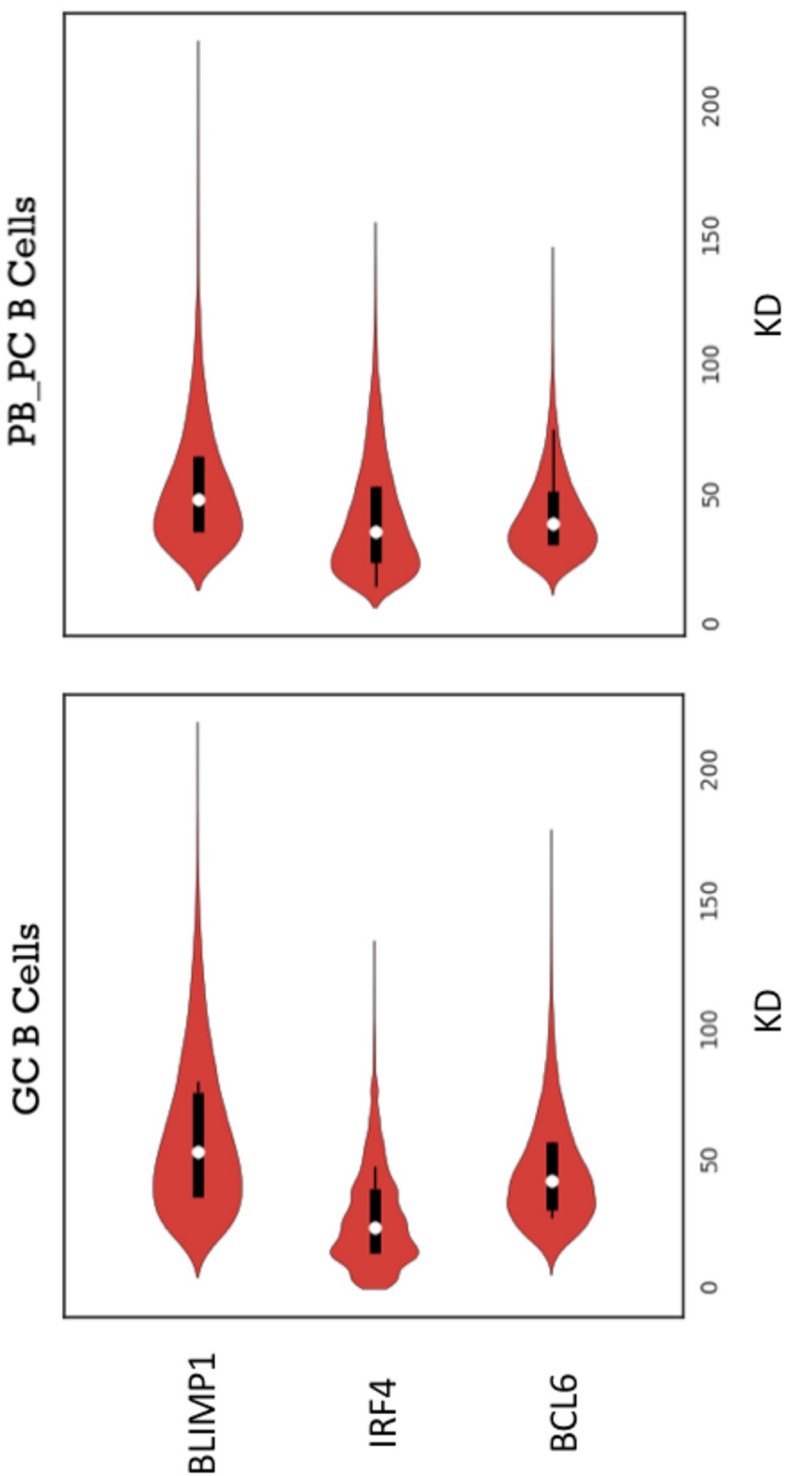

**Fig 3. Model-to-model distributions of KD for GC and PB_PC stages and the three genes, BCL6, IRF4, BLIMP1.**
Model (4)–(6) was simulated with parameter values from Tables 2–5, version I. The violin plots show the shapes of the distributions, median value, interquartile range and 1.5x interquartile range of the KD values.

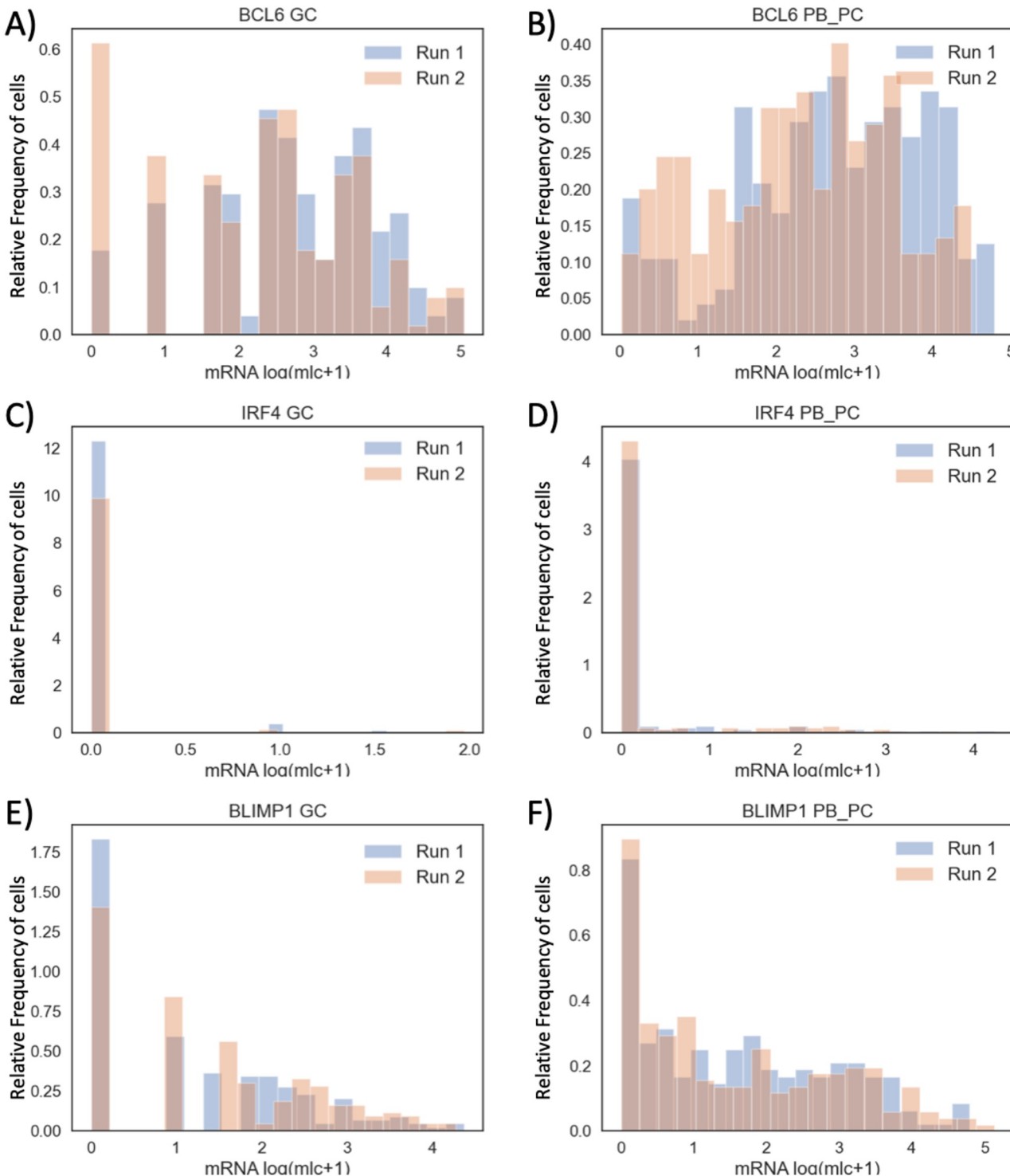

**Fig 4. Histograms of two model-generated mRNA counts of BCL6, IRF4 and BLIMP1 at GC and PB_PC stages with the highest KD.** The subgraphs A, C, E (resp., B, D, F) represent the relative frequency of cells (y-axis) for log2 (molecule+1) transformed values of BCL6, IRF4 and BLIMP1 (x-axis) at GC (resp., PB_PC) stage. Parameters from Tables 2–5, version I.

Indeed, if IRF4 autoactivation reaction is not efficient enough, there are not enough IRF4 molecules to affect BCL6 and BLIMP1 activity at PB_PC stage. Because IRF4 activity is only impacted by its autoactivation loop, we first modulated values of the parameter related to this reaction ($\theta_{2,2}$). During preliminary tests, we noticed that this reaction is crucial for the transition from GC towards PB_PC stage and that when interaction parameter $\theta_{2,2}$ and transcription rate $s_{0,IRF4}$ have low absolute values then the system cannot reach PB_PC stage, even after application of the stimuli. It can be explained by the insufficient amount of IRF4 molecules produced (S3C and S3D Fig). On the other hand, when parameters $\theta_{2,2}$ and $s_{0,IRF4}$ have high values, model (4)–(6) transitions from GC towards PB_PC stage even before application of stimuli, exhibiting an abnormal behavior.

After comparison of the stochastic system (4)–(6) outputs for a range of different $\theta_{2,2}$ and $s_{0,IRF4}$ values (described in Table 6), we selected the parameter set for which model (4)–(6) correctly fits the IRF4 experimental data at both GC and PB_PC stages. Such model-derived SC pattern is obtained using the values ($\theta_{2,2} = 11$ and $s_{0,IRF4} = 2.1$ molecule.h$^{-1}$).

We additionally performed simulations to improve the quality of the fitting of BLIMP1 and BCL6 distributions by testing parameters that are directly responsible for the balance between BLIMP1 and BCL6, such as interaction parameters $\theta_{1,3}$, $\theta_{3,1}$ and $\theta_{2,3}$. We also tested parameters which can influence BCL6 and BLIMP1 indirectly, such as transcription rates ($s_{0,BCL6}$ and $s_{0,BLIMP1}$), and degradation rates of mRNA ($d_{0,BCL6}$, $d_{0,IRF4}$ and $d_{0,BLIMP1}$).

After comparison of the stochastic system (4)–(6) outputs, we selected the parameters which allow the model to have a qualitative fit of the experimental data for all nodes at GC and PB_PC stages (Fig 5, and Tables 2–5, version III). For this tuned parameter set, we see that the model (4)–(6) can have a good qualitative fitting of experimental data for all nodes. Results also show that for this parameter set (version III), the stochastic model (4)–(6) fits SC data at the GC stage for BCL6 (Fig 5A). The model-derived empirical distribution of BLIMP1 was capable of showing overlap with experimental data at the PB_PC stage (Fig 5F), but it overestimated the number of BLIMP1 mRNA molecules at the GC stage (Fig 5E).

The current parameter set (Tables 2–5, version III) has difficulties to correctly evaluate the number of zero values. The model (4)–(6) tends to overestimate the number of BCL6 mRNA molecules at PB_PC stage, as well as the number of IRF4 mRNA molecules at GC stage and number of BLIMP1 mRNA molecules at GC stage (Fig 5). Nevertheless, this parameter set allowed the model to generate SC data with a similar level of magnitude of the amount of mRNA as experimentally observed.

## Discussion

In this work, we applied a particular class of stochastic models combining deterministic dynamics and random jumps to the simulation of SC data from two stages of B cell differentiation in germinal centers.

We first defined a reduced model (9) whose dynamics were compared to the ones of the kinetic model (1)–(3) and we established an initial parameter value for the key parameter $k_{on,init,IRF4}$. We then showed that for a given parameter set (Table 2–5, Version I), the reduced model (9) admits two steady states. Secondly, we evaluated the effect of stochasticity on multiple independent generations of the number of mRNA molecules by the stochastic model (4)–(6) and we confirmed that for the same parameter set there is no noticeable difference between each model-generated outputs for BCL6-IRF4-BLIMP1 GRNs (Fig 4). These results allow performing a combined parameter screening with the confidence that for each candidate parameter set, the algorithm needs to perform only one run of the model (4)–(6). Lastly, we showed that the model (4)–(6) can simulate distributions of the number of mRNA

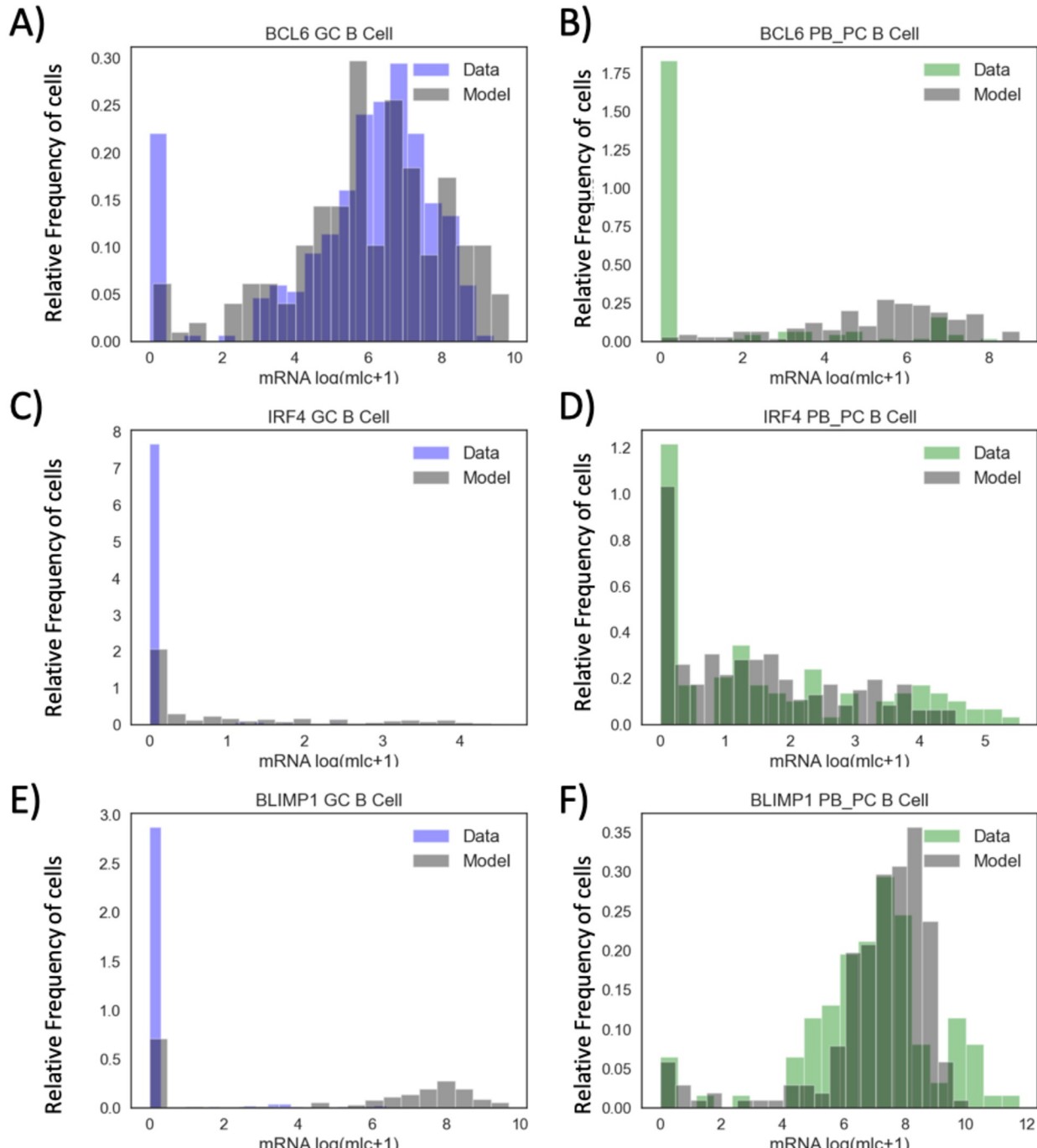

**Fig 5. Histograms of model-generated and experimental mRNA counts of BCL6, IRF4, BLIMP1 at GC and PB_PC stages.** The subgraphs A, C, E (resp., B, D, F) represent the relative frequency of cells (y-axis) for log2 (molecule+1) transformed values of BCL6, IRF4 and BLIMP1 (x-axis) at GC (resp., PB_PC) stage, compared between the model estimations at GC or PB_PC stage (grey) vs the experimental data from GC (blue) or PB_PC (green) B cells. Simulation of 200 single cells were used based on the parameter set, selected after semi-automatized parameter screening (see Tables 2–5, version III). Performed based on the dataset from Milpied et al. [30].

molecules for BCL6, IRF4, BLIMP1 at GC and PB_PC stages with the same order of magnitude as experimental data. However, as future scope of this work, a few strategies to improve the final parameter value set (Tables 2–5, version III) can be investigated.

Since in BCL6-IRF4-BLIMP1 GRN, IRF4 activity depends only on its autoactivation reaction, we have only succeeded, by writing the reduced model (9) in terms of the kinetic model (1)–(3), in estimating the value of $k_{\mathrm{on,init,IRF4}}$. It would be advantageous to additionally estimate the values of $k_{\mathrm{on,init,BCL6}}$ and $k_{\mathrm{on,init,BLIMP1}}$, using the same logic. However, because BLIMP1 depends on BLIMP1, IRF4 and BCL6 (see Eq (1)) and BCL6 depends on both IRF4 and BLIMP1 (Eq (2)), the rewriting of system (4)–(6) in terms of (1)–(3) would require additional calculations and simplifications.

The effect of mutual repression between BCL6 and BLIMP1 could be evaluated by performing a more extensive parameter value search. The current parameter value set (Tables 2–5, version III) makes model (4)–(6) overestimate the number of mRNA molecules of BLIMP1 at GC stage. Increasing BCL6 repression of BLIMP1 could potentially decrease the quantity of BLIMP1 at the GC stage.

The effect of the duration of the BCR and CD40 stimuli on the differentiation from GC B cells towards PB_PC could be investigated. Multiscale modeling of GCs performed by Tejero et al. [39] showed that CD40 signalling in combination with the asymmetric division of B cells results in a switch from memory B cells to plasmablasts. It would be relevant to evaluate a possible application of the stochastic model to study the effect of combined CD40 and BCR signaling with different intensities and durations at the SC level.

Additionally, one can evaluate the impact of including additional genes into the BCL6-IRF4-BLIMP1 GRN on the quality of data fitting by the stochastic model. One of the possible candidates to incorporation in the GRN is PAX5, which plays an important role in directing lymphoid progenitors towards B cell development [40]. PAX5 positively regulates IRF8 and BACH2, which indirectly positively regulate IRF4 and negatively regulate BLIMP1 at an early stage of B cell differentiation. During further development, BLIMP1 starts to repress PAX5, consequently decreasing the expression of IRF8 and BACH2. The correct orchestration of PAX5-IRF8-BACH2 during B cell differentiation is important for the successful differentiation towards antibody producing cells (PB_PC), while its malfunction can cause aberration in GC B cell development [41].

CD40 stimulation of B cells initiates NF-$\kappa B$ signaling which is associated with cellular proliferation. In B cells, NF-$\kappa B$ activates IRF4, negatively regulates BACH2, what leads to positive regulation of BLIMP1 and consecutive repression of BCL6 [4, 34].

Another important transcription factor in GC development is MYC, which regulates B cell proliferation [42] and the DZ B cell phenotype [43]. MYC indirectly activates the histone methyltransferase enhancer of zeste homologue 2 (EZH2), which is responsible for the repression of IRF4 and BLIMP1 [44–47].

The transcription factors mentioned above are present in the SC RT-qPCR dataset from Milpied et al. [30] that we used and could be used to extend the current GRN. Inclusion of additional transcription factors may have both positive and negative effects on the application of model (4)–(6). On one side, it can increase the computational time and the number of parameters required for simulating System (4)–(6). On the other side, because the inclusion of transcription factors can more precisely describe the biological system it could improve the quality of the fitting. However, any inclusion of new nodes to GRN should be carefully evaluated and only essential transcription factors should be added. For instance, there are no advantages in adding a transcrption factor that would only have one downstream output. As an example, MYC activates E2F1 and further activates EZH2. For this reason, incorporation of the chain MYC-E2F1-EZH2 should have a similar outcome, as the incorporation of the

simplified MYC-EZH2 reaction. This is expected because in the modeling, intermediate elements of one-to-one redundant reactions can be omitted without significant changes in the quality of the simulations.

To further continue our study, we could also use SC RNA-seq dataset from Milpied et. al [30]. The authors have produced SC RNA-seq dataset from GC B cells and analysed the similarities between SC RNA-seq and SC RT-qPCR dataset. Even though the gene-gene correlation levels were lower in SC RNA-seq compared to SC RT-qPCR, SC RNA-seq analysis confirmed the observation obtained by SC RT-qPCR [30]. From the stochastic modeling perspective, combining the data from SC RT-qPCR and SC RNA-seq should improve our understanding of the SC dataset variability and the quality of the fitting.

To summarise, the stochastic model (4)–(6) is capable of qualitatively simulating and depicting the stochasticity of experimental SC gene expression data of human B cell differentiation at the GC and PB_PC stages using a GRN made of three-key genes (BCL6, IRF4, BLIMP1). These results are encouraging, and suggest that our model may be used to test the different B cell exits from GC. Future steps may include testing of the model (4)–(6) on alternative SC datasets [48–50] and investigating the malignant formations, by evaluating differences of the associated GRN compared to the normal B cell differentiation from GC towards PB_PC.

## Supporting information

**S1 Fig. Scheme of application of the stimuli $Q_s$, where $s \in \{BCR, CD40\}$.**
(TIF)

**S2 Fig. Absence of bistability in model (9).**
(TIF)

**S3 Fig. Histograms of model-generated and experimental mRNA counts of BCL6, IRF4, BLIMP1 at GC and PB_PC stages.**
(TIF)

**S1 File. Modeling.** This file introduces the methodology for reducing the stochastic model and estimating the initial activation rate.
(PDF)

**S1 Table. Parameters of system (9) with values accordingly to Bonnaffoux et al. [32].**
(PDF)

## Acknowledgments

We thank Arnaud Bonnaffoux and Matteo Bouvier for their help with the WASABI framework and their critical reading of the manuscript. We thank the computational center of IN2P3 (Villeurbanne/France), especially Gino Marchetti and Renaud Vernet. We also thank Aurelien Pélissier and Elias Ventre for their scientific discussions.

## Author Contributions

**Conceptualization:** Alexey Koshkin, Olivier Gandrillon, Fabien Crauste.

**Data curation:** Alexey Koshkin.

**Formal analysis:** Alexey Koshkin, Fabien Crauste.

**Funding acquisition:** Olivier Gandrillon, Fabien Crauste.

**Investigation:** Alexey Koshkin, Ulysse Herbach, Olivier Gandrillon, Fabien Crauste.

**Methodology:** Alexey Koshkin, Ulysse Herbach, María Rodríguez Martínez, Olivier Gandrillon, Fabien Crauste.

**Project administration:** Olivier Gandrillon, Fabien Crauste.

**Resources:** Olivier Gandrillon, Fabien Crauste.

**Supervision:** María Rodríguez Martínez, Olivier Gandrillon, Fabien Crauste.

**Validation:** María Rodríguez Martínez, Olivier Gandrillon, Fabien Crauste.

**Visualization:** Alexey Koshkin, Olivier Gandrillon.

**Writing – original draft:** Alexey Koshkin.

**Writing – review & editing:** Ulysse Herbach, María Rodríguez Martínez, Olivier Gandrillon, Fabien Crauste.

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
