## [Decision Letter · Decision Letter 0]

20 Dec 2023

PONE-D-23-18580Stochastic modeling of a gene regulatory network driving B cell development in germinal centersPLOS ONE

Dear Dr. Crauste,

Thank you for submitting your manuscript to PLOS ONE. After careful consideration, we feel that it has merit but does not fully meet PLOS ONE’s publication criteria as it currently stands. Therefore, we invite you to submit a revised version of the manuscript that addresses the points raised during the review process.

Your manuscript, "Stochastic modeling of a gene regulatory network driving B cell development in germinal centers" (PONE-D-23-18580), has been assessed by our reviewers. Although it is of interest, we are unable to consider it for publication in its current form. The reviewers have raised a number of mains points that we believe would improve the manuscript and may allow a revised version to be published in PLOS ONE.

We look forward to receiving your revised manuscript.

Kind regards,

Nihad A.M Al-Rashedi

Academic Editor

PLOS ONE

Journal Requirements:

“AK, UH, MRM, OG, FC. This work was supported by the COSMIC grant (www.cosmic-h2020.eu) which has received funding from European Union’s Horizon 2020 research and innovation program under the Marie Sklodowska-Curie grant agreement no. 765158.”

3. PLOS requires an ORCID iD for the corresponding author in Editorial Manager on papers submitted after December 6th, 2016. Please ensure that you have an ORCID iD and that it is validated in Editorial Manager. To do this, go to ‘Update my Information’ (in the upper left-hand corner of the main menu), and click on the Fetch/Validate link next to the ORCID field. This will take you to the ORCID site and allow you to create a new iD or authenticate a pre-existing iD in Editorial Manager. Please see the following video for instructions on linking an ORCID iD to your Editorial Manager account: https://www.youtube.com/watch?v=_xcclfuvtxQ.

4. Please update your submission to use the PLOS LaTeX template. The template and more information on our requirements for LaTeX submissions can be found at http://journals.plos.org/plosone/s/latex.

5. We notice that your supplementary figures are uploaded with the file type 'Figure'. Please amend the file type to 'Supporting Information'. Please ensure that each Supporting Information file has a legend listed in the manuscript after the references list.

Reviewers' comments:

Reviewer's Responses to Questions

**Comments to the Author**

1. Is the manuscript technically sound, and do the data support the conclusions?

Reviewer #1: Yes

Reviewer #2: Yes

2. Has the statistical analysis been performed appropriately and rigorously? 

Reviewer #1: I Don't Know

Reviewer #2: Yes

3. Have the authors made all data underlying the findings in their manuscript fully available?

Reviewer #1: Yes

Reviewer #2: Yes

4. Is the manuscript presented in an intelligible fashion and written in standard English?

Reviewer #1: Yes

Reviewer #2: Yes

5. Review Comments to the Author

Reviewer #1: The manuscript by Alexey Koshkin et al. deals with the investigation of the gene regulatory networks (GRNs) associated with germinal center (GC) cell development and differentiation based on public available single-cell (SC) transcriptomic data. Including three key gene regulators (BCL6, IRF4, BLIMP1), influenced by two external stimuli signals (surface receptors BCR and CD40), a model was established that qualitatively recapitulates mRNA distributions corresponding to GC and plasmablast stages of B cell differentiation, which can be used in validating the GRN in physiological and pathophysiological conditions.

The manuscript is written well and meets almost all criterias for publishing in PlosOne:

Comments:

The validation (testing) of the model on a test-data set (for example the sc-RNA data set from the same sample source but also an external data set would highly improve the quality of the paper.

Reviewer #2: In this paper the authors use stochastic modeling of a gene regulatory network to fit single cell expression data related to B cell differentiation in germinal centers. This is an important problem which can lead to better understanding of malignancies in B cells. The work is technically sound, and the paper is well-written with the mathematical descriptions and the figures doing a good job of clearly presenting the work to the readers. I would like to point out some revisions that are still needed.

- The benefit of using a stochastic model over a kinetic model, as also the improvements of using Version III over Version II are not fully clear without including a figure like Figure 2 each for the latter two parameter tune cases. For example, in version III PB_PC improves for two genes, however the GC performance becomes worse. So plots showing the mRNA counts like in Figure 2 would show how close the total model predictions are to the observed values.

- Some of the parameters depicted in 4-6 are not present later, for example beta_i, gamma, k^min, k^max. It needs to be clarified how they are replaced (e.g. maybe by k_init) or not used anymore.

- In Pages 6 and 7, line 146-154, the ranges for some parameters are not given, e.g. theta_{1,1}, H_{1,1}, etc. In line 152, aren't there 6 H_{j,i} interactions and 11 theta_{i,j} interactions? The values in Line 153 do not match, so that section needs to corrected so that the number of parameter combinations adds up.

- Why is CD40 stimuli upto 61 instead of 60?

- In Figures 4, 5, S3, some further details about the y axis scale are needed for the reader.

- In Figure 1 caption, k_{off,i} is not shown to be dependent on P_j and theta_ji in the text, only k_{on,i}, so those should match. Also autoactivation loop for BCL6 needs to be mentioned, as it is mentioned just for IRF4.

- Error in Line 363 (antibody producing, not antigen producing) should be corrected.

6. PLOS authors have the option to publish the peer review history of their article (what does this mean?). If published, this will include your full peer review and any attached files.

Reviewer #1: No

Reviewer #2: No

---

## [Author Response · Author response to Decision Letter 0]

20 Feb 2024

All our responses to reviewers comments are detailed in the Response-to-Reviewers.pdf file attached to this submission.

---

## [Decision Letter · Decision Letter 1]

11 Mar 2024

Stochastic modeling of a gene regulatory network driving B cell development in germinal centers

PONE-D-23-18580R1

Dear Dr. Crauste,

We’re pleased to inform you that your manuscript has been judged scientifically suitable for publication and will be formally accepted for publication once it meets all outstanding technical requirements.

Kind regards,

Nihad A.M Al-Rashedi

Academic Editor

PLOS ONE

Additional Editor Comments (optional):

Reviewers' comments:

Reviewer's Responses to Questions

**Comments to the Author**

1. If the authors have adequately addressed your comments raised in a previous round of review and you feel that this manuscript is now acceptable for publication, you may indicate that here to bypass the “Comments to the Author” section, enter your conflict of interest statement in the “Confidential to Editor” section, and submit your "Accept" recommendation.

Reviewer #1: All comments have been addressed

Reviewer #2: All comments have been addressed

2. Is the manuscript technically sound, and do the data support the conclusions?

Reviewer #1: Yes

Reviewer #2: (No Response)

3. Has the statistical analysis been performed appropriately and rigorously? 

Reviewer #1: Yes

Reviewer #2: (No Response)

4. Have the authors made all data underlying the findings in their manuscript fully available?

Reviewer #1: Yes

Reviewer #2: (No Response)

5. Is the manuscript presented in an intelligible fashion and written in standard English?

Reviewer #1: Yes

Reviewer #2: (No Response)

6. Review Comments to the Author

Reviewer #1: The manuscript by Alexey Koshkin et al. deals with the investigation of the gene regulatory networks (GRNs) associated with germinal center (GC) cell development and differentiation based on public available single-cell (SC) transcriptomic data. Including three key gene regulators (BCL6, IRF4, BLIMP1), influenced by two external stimuli signals (surface receptors BCR and CD40), a model was established that qualitatively recapitulates mRNA distributions corresponding to GC and plasmablast stages of B cell differentiation, which can be used in validating the GRN in physiological and pathophysiological conditions.

The manuscript is written well and meets almost all criterias for publishing in PlosOne. My concern was

taken into account and the response was accepted by the reviewer. Therefore, the paper is fit for publication in "Plos ONE"

Reviewer #2: (No Response)

7. PLOS authors have the option to publish the peer review history of their article (what does this mean?). If published, this will include your full peer review and any attached files.

Reviewer #1: No

Reviewer #2: **Yes: **Gourab Ghosh Roy

---

## [Editor Report · Acceptance letter]

13 Mar 2024

PONE-D-23-18580R1 

PLOS ONE

Dear Dr. Crauste, 

I'm pleased to inform you that your manuscript has been deemed suitable for publication in PLOS ONE. Congratulations! Your manuscript is now being handed over to our production team.

Kind regards, 

on behalf of

Dr. Nihad A.M Al-Rashedi 

Academic Editor

PLOS ONE